# A Characterization of Multipliers of the Herz Algebra

Hans G. Feichtinger [1,2]

1   Faculty of Mathematics, University of Vienna, NuHAG Oskar-Morgenstern-Platz 1, 1090 Vienna, Austria; hans.feichtinger@univie.ac.at

2   ARI (Acoustic Research Institute, OEAW), Wohllebengasse 12-14, 1040 Vienna, Austria

**Abstract:** For the characterization of multipliers of $L^p(\mathbb{R}^d)$ or more generally, of $L^p(G)$ for some locally compact Abelian group $G$, the so-called Figa-Talamanca–Herz algebra $A_p(G)$ plays an important role. Following Larsen's book, we describe multipliers as bounded linear operators that commute with translations. The main result of this paper is the characterization of the multipliers of $A_p(G)$. In fact, we demonstrate that it coincides with the space of multipliers of $(L^p(G), \|\cdot\|_p)$. Given a multiplier $T$ of $(A_p(G), \|\cdot\|_{A_p(G)})$ and using the embedding $(A_p(G), \|\cdot\|_{A_p(G)}) \hookrightarrow (C_0(G), \|\cdot\|_\infty)$, the linear functional $f \mapsto [T(f)(0)]$ is bounded, and $T$ can be written as a moving average for some element in the dual $PM_p(G)$ of $(A_p(G), \|\cdot\|_{A_p(G)})$. A key step for this identification is another elementary fact: showing that the multipliers from $(L^p(G), \|\cdot\|_p)$ to $(C_0(G), \|\cdot\|_\infty)$ are exactly the convolution operators with kernels in $(L^q(G), \|\cdot\|_q)$ for $1 < p < \infty$ and $1/p + 1/q = 1$. The proofs make use of the space of mild distributions, which is the dual of the Segal algebra $(S_0(G), \|\cdot\|_{S_0})$, and the fact that multipliers $T$ from $S_0(G)$ to $S_0'(G)$ are convolution operators of the form $T : f \mapsto \sigma * f$ for some uniquely determined $\sigma \in S_0'$. This setting also allows us to switch from the description of these multipliers as convolution operators (by suitable pseudomeasures) to their description as Fourier multipliers, using the extended Fourier transform in the setting of $(S_0'(G), \|\cdot\|_{S_0'})$. The approach presented here extends to other function spaces, but a more detailed discussion is left to future publications.

**Keywords:** Fourier multipliers; Herz algebra; Herz–Figa-Talamanca space; pseudomeasures; quasimeasures; mild distributions; Feichtinger algebra





## 1. Introduction

According to Larsen ([1]), whenever one has two translation invariant Banach spaces of functions $(B^1, \|\cdot\|^{(1)})$ and $(B^2, \|\cdot\|^{(2)})$ over some LCA (locally compact Abelian) group $G$ (following a general suggestion of H. Reiter, a theorem should not be trivial when specialized to the Euclidean case, i.e., to the case $G = \mathbb{R}^d$), it is natural to ask for a characterization of all the "multipliers", i.e., bounded linear operators that commute with translations. The translation operators $T_x, x \in G$ are defined by $T_x f(z) = f(z - x)$ for ordinary functions, and are extended to spaces of distributions by duality.

Given two Banach spaces $(B^1, \|\cdot\|^{(1)})$ and $(B^2, \|\cdot\|^{(2)})$, we write

$$H_G(B^1, B^2) := \{T \in L(B^1, B^2) \mid T \circ T_x = T_x \circ T, \forall x \in G\}$$

where

$$L(B^1, B^2) := \{T : (B^1, \|\cdot\|^{(1)}) \to (B^2, \|\cdot\|^{(2)}), \text{ bounded and linear}\}$$

Clearly, $L(B^1, B^2)$ is a Banach space with the usual operator norm

$$\||T|\|_{B^1 \to B^2} = \sup_{\|f\|_{B^1} \le 1} \{\|Tf\|_{B^2}\}$$

and the convention of writing $\||T\||_B$ for the case $B^1 = B = B^2$.

In order to make the definition of $H_G(B^1, B^2) \hookrightarrow L(B^1, B^2)$ (as a closed subspace) meaningful, we assume of course that both spaces $(B^1, \|\cdot\|^{(1)})$ and $(B^2, \|\cdot\|^{(2)})$ are translation invariant, i.e., that $T_x B^1 \subseteq B^1$ and $T_x B^2 \subseteq B^2$ for any $x \in G$. As usual, we write $L(B)$ instead of $L(B, B)$ and, correspondingly, $H_G(B)$ for $H_G(B^1, B^2)$ for the case $B^1 = B = B^2$. For $B = L^p$, the symbol $CV_p(G)$ is found in the literature (for "convolutors") instead of $H_G(L^p)$ (see [2,3]).

The goal of this paper is to show that elementary arguments with the potential for significant generalizations (to be discussed elsewhere) allow the identification of the multipliers of the so-called Herz-algebra $(A_p(G), \|\cdot\|_{A_p(G)})$ with its dual space. By establishing the connection to multipliers of $(L^p(G), \|\cdot\|_p)$, we show that these two multiplier spaces coincide, thus providing an alternative approach to the well-known characterization of the convolutors of $(L^p(G), \|\cdot\|_p)$. In the context of mild distributions (which work well for general LCA groups), their Fourier transforms are simply the Fourier multipliers of $(L^p(G), \|\cdot\|_p)$.

The word "multiplier" refers to the typical description of such operators as "Fourier multipliers", i.e., as pointwise multipliers on the Fourier transform side:

$$\mathcal{F}(T(f)) = h \cdot \widehat{f} \quad \text{or} \quad T(f) = \mathcal{F}^{-1}[h \cdot \mathcal{F}(f)]. \tag{1}$$

Let us mention that one must be careful with such a description, because it requires that both sides of the equation are interpreted properly: The Fourier transform of $f$ as well as that of $T(f)$ have to be meaningful, but the pointwise product must also be well defined.

Although the general setting for such a question are Banach spaces of distributions over LCA (locally compact Abelian) groups $G$, we will illustrate this problem mostly in the (typical) Euclidean context ($G = \mathbb{R}^d$), and restrict our attention to the unweighted case.

A natural starting point for such a discussion is the analysis of $H_G(L^p)$, for $1 \leq p < \infty$ over $G = \mathbb{R}^d$. Here, the case $p = \infty$ is excluded for good reasons, because there are "exotic" operators in $H_G(L^\infty(\mathbb{R}^d))$ that cannot be represented as convolution operators, not even in the distributional setting.

Even if one restricts the attention to the space $C_b(\mathbb{R}^d)$ of bounded, continuous functions, i.e., if one replaces the problem by considering $H_G(C_b(\mathbb{R}^d))$, this problem persists, as it is explained in a series of papers by I. Sandberg (mentioned in [4]; see [5–7]). One can argue that the main reason is the fact that the closure of the test functions ($\mathcal{S}(\mathbb{R}^d)$ or $S_0(\mathbb{R}^d)$) in the given space is strictly contained in $C_b(\mathbb{R}^d)$, and thus the Hahn–Banach theorem allows the creation of non-trivial exotic, translation invariant functionals that vanish on the test functions.

We start the discussion with the few cases where a full characterization of $H_G(L^p)$ can be given, namely, the cases $p = 1$ and $p = 2$. Following the presentation of [1] (or with different notations [2]), they read as follows:

**Theorem 1** (Wendel's Theorem). *There is a natural isomorphism between $H_G(L^1(\mathbb{R}^d))$ and $(M_b(\mathbb{R}^d), \|\cdot\|_{M_b})$, which identifies the space of multipliers with the space of convolution operators by bounded measures, i.e.,*

$$T \in H_G(L^1(\mathbb{R}^d)) \leftrightarrow T(f) = \mu * f, \quad f \in L^1(\mathbb{R}^d)$$

*for a uniquely determined $\mu \in M_b(\mathbb{R}^d)$. Moreover, we have*

$$\||T\||_{L^1 \to L^1} = \|\mu\|_{M_b}.$$

*Correspondingly, we have $\mathcal{F}(T(f)) = \widehat{\mu} \cdot \widehat{f}$, for $f \in L^1(\mathbb{R}^d)$, i.e., the operator is described as a pointwise multiplier of $\widehat{f} \in C_0(\mathbb{R}^d)$ with the (uniformly continuous and bounded) Fourier–Stieltjes transform $\widehat{\mu} \in C_{ub}(\mathbb{R}^d)$.*

In the theorem above, the space $M_b(\mathbb{R}^d)$ is understood as the Banach space of bounded, regular Borel measures on $\mathbb{R}^d$. They correspond to bounded linear functionals on $\big(C_0(\mathbb{R}^d), \|\cdot\|_\infty\big)$ (the space of continuous complex-valued functions on $\mathbb{R}^d$, vanishing at infinity) via

$$\mu(f) := \int_{\mathbb{R}^d} f(t)\,d\mu(t), \quad f \in C_0(\mathbb{R}^d).$$

This convention (fully developed in [4]) is justified by the Riesz Representation theorem (in its locally compact version). Moreover, the natural norm (the *total variation norm*) coincides with the norm of the corresponding functional. Since $C_c(\mathbb{R}^d) := \{k \in C_b(\mathbb{R}^d) \mid \operatorname{supp}(k) \text{ compact}\}$ is dense in $\big(C_0(\mathbb{R}^d), \|\cdot\|_\infty\big)$, this means that for $\mu \in M_b(\mathbb{R}^d)$, one has

$$\|\mu\|_{\mathrm{TV}} = \sup_{f \in C_c(\mathbb{R}^d), \|f\|_\infty \leq 1} |\mu(f)|.$$

Another clear-cut case appears for $p = 2$, as a consequence of *Plancherel's Theorem*. Once the usual Fourier transform defined on $L^1(\mathbb{R}^d)$ has been extended from $L^1 \cap L^2(\mathbb{R}^d)$ to a *unitary automorphism* of the Hilbert space $\mathcal{H} = \big(L^2(\mathbb{R}^d), \|\cdot\|_2\big)$, we can formulate the characterization of $H_G(L^2(\mathbb{R}^d))$ as follows:

**Theorem 2.** *There is a natural identification of $H_G(L^2(\mathbb{R}^d))$ with $\big(L^\infty(\mathbb{R}^d), \|\cdot\|_\infty\big)$, the space of essentially bounded (equivalence classes of) measurable functions on $\mathbb{R}^d = \widehat{\mathbb{R}^d}$, via*

$$\mathcal{F}(T(f)) = h \cdot \mathcal{F}(f), \quad f \in L^2(\mathbb{R}^d). \tag{2}$$

*This correspondence between $T$ and $h$ defines an isometric isomorphism between $H_G(L^2(\mathbb{R}^d))$ and the Banach space of all pointwise multipliers of $\big(L^2(\mathbb{R}^d), \|\cdot\|_2\big)$; hence, we have*

$$\|\!|T|\!\|_{L^2 \to L^2} = \|h\|_\infty.$$

**Remark 1.** *The two characterizations correspond quite well to the two concepts used for the description of TILS ( time-invariant linear systems) in introductory engineering courses on "systems theory". It appears as Theorem 4.1.1 in [1], or as Theorem 1 in Section 1.3 of [2] for general LCA groups. The relevance of this viewpoint for applications is also communicated in [8].*

**Remark 2.** *In the context of (mild or) tempered distributions (see [9] and or [10] respectively [11]), one can form $PM(\mathbb{R}^d) := \mathcal{F}^{-1}(L^\infty(\mathbb{R}^d))$ and call this (by transfer of the norm) the space of pseudomeasures. This space plays an important role for spectral analysis (see the book [12] by J. Benedetto). Note that the natural pointwise multiplication structure (which is equivalent to the Banach algebra properties of $H_G(L^2(\mathbb{R}^d))$ obtained via composition of operators) corresponds to a natural form of convolution for pseudomeasures. However, one must be warned that even in the case of regular pseudomeasures, corresponding to continuous bounded functions on $\mathbb{R}^d$, this does not mean that convolution is meaningful in the pointwise sense. The standard example are the chirp functions $\psi_\alpha(t) = exp(i\alpha t^2)$. The general version of such a chirp or linear frequency sweep is obtained from $\psi_1$ by dilation. This particular case has the remarkable property of being an eigenvector of the Fourier transform (the eigenvalue depends on the normalization of the Fourier transform). Since $\psi_1 \in C_b(\mathbb{R}) \subset L^\infty(\mathbb{R})$, these function represent pseudomeasures.*

**Remark 3.** *What is called the convolution kernel $\mu$ in Wendel's Theorem is described as the "impulse response" of the system $T$, while the Fourier multiplier $h$ is called the transfer function of $T$. It describes essentially the resonance behavior of the system, i.e., how the joint eigenvectors of such systems, namely the pure frequencies, are amplified (or damped) by the system. However, there are many problems if one tries to put such statements into a correct mathematical setting, since, a priori, a bounded operator on any of the spaces $\big(L^p(\mathbb{R}^d), \|\cdot\|_p\big)$ with $p < \infty$ cannot have pure frequencies of the form $\chi_s(t) = exp(2\pi i s t)$ as eigenvectors in the strict sense.*

**Remark 4.** *Let us mention that the proof of Wendel's Theorem is similar to the engineering approach. It starts by observing that the convolution algebra $\left(L^1(\mathbb{R}^d), \|\cdot\|_1\right)$ has bounded approximate units or Dirac sequences of norm one, obtained typically by applying the isometric compression operator $\mathrm{St}_\rho$, given by*

$$\mathrm{St}_\rho(g)(x) = \rho^{-d} g(x/\rho), \quad \rho > 0$$

*to any function with $\int_{\mathbb{R}^d} g(x) dx = \widehat{g}(0) = 1$. Applying an operator $T \in \mathbf{H}_G(L^1(\mathbb{R}^d))$, one obtains a bounded family in $\left(L^1(\mathbb{R}^d), \|\cdot\|_1\right) \hookrightarrow (M_b(\mathbb{R}^d), \|\cdot\|_{M_b})$ (isometrically), which allows one select a $w^*$-convergent subsequence with some limit $\mu \in M_b(\mathbb{R}^d)$. Then, it is shown that $T(f) = \mu * f$ for $f \in L^1(\mathbb{R}^d)$.*

*The Fourier transform intertwines $\mathrm{St}_\rho$ with its adjoint operator, given by $\mathrm{D}_\rho h(x) = h(\rho x)$. In other words, $\mathcal{F}(\mathrm{St}_\rho f) = \mathrm{D}_\rho(\widehat{f})$, $f \in L^1(\mathbb{R}^d)$. The family $(\mathrm{D}_\rho h)_{\rho \to 0}$ defines a bounded approximate unit in the Fourier algebra $\left(\mathcal{F}L^1(\mathbb{R}^d), \|\cdot\|_{\mathcal{F}L^1}\right)$.*

**Remark 5.** *Engineering students are then told that "by experience" one observes that (somehow!) there is a limit $\mu = \lim_{\rho \to 0} T(\mathrm{St}_\rho g)$, which will be called the impulse response, thus suggesting that one has, using the fact that $T$ commutes also with convolutions,*

$$T(f) = \lim_{\rho \to 0} T(\mathrm{St}_\rho g * f) = [\lim_{\rho \to 0} T(\mathrm{St}_\rho g)] * f = \mu * f. \tag{3}$$

*Usually, $g$ is chosen to be either the boxcar function $\mathbf{1}_{[-1/2,1/2]}$, or a standard Gaussian (which is invariant under the Fourier transform).*

**Remark 6.** *The discussed two cases ($p = 1, 2$) indicate that any $T \in \mathbf{H}_G(L^1(\mathbb{R}^d))$ extends to a multiplier for $\left(L^p(\mathbb{R}^d), \|\cdot\|_p\right)$, in particular for $p = 2$ (with transfer function $h = \widehat{\mu}$), since we have for $1 \leq p \leq \infty$ (see Remark 8 below):*

$$\|\mu * f\|_{L^p(\mathbb{R}^d)} \leq \|\mu\|_{M_b} \|f\|_{L^p(\mathbb{R}^d)}, \quad f \in L^p(\mathbb{R}^d). \tag{4}$$

*The case $p = \infty$ is obvious since $L^\infty(\mathbb{R}^d)$ is the dual space of $L^1(\mathbb{R}^d)$, and for $1 \leq p < \infty$ the spaces $\left(L^p(\mathbb{R}^d), \|\cdot\|_p\right)$ are homogeneous Banach spaces; thus, one can obtain the estimate (14) using the approach outlined in [13] (cf. [2], Chap.1.2).*

**Remark 7.** *Most classical books on Fourier analysis (notably [14] respectively [15]) during the last century give the impression that integration theory is the basis for harmonic analysis. After all, the Banach space $\left(L^1(\mathbb{R}^d), \|\cdot\|_1\right)$ (or $\left(L^1(G), \|\cdot\|_1\right)$, for a LCA group) appears as the natural domain for the Fourier transform, viewed as an integral transform. In fact, even the pointwise definition of convolution appears to require (via Fubini's theorem) the integrability of the factors. From there, it is quite natural to study $\left(L^1(\mathbb{R}^d), \|\cdot\|_1\right)$ as a Banach algebra with respect to convolution, and to derive the convolution theorem, in which convolution is turned into pointwise multiplication by the Fourier transform. This is one of the key properties of the Fourier transform, along with the fact that it defines a unitary automorphism of $\left(L^2(\mathbb{R}^d), \|\cdot\|_2\right)$ (by Plancherel's theorem).*

*The key result of [4] provides an alternative approach (which has been tested in various courses in the last decades by the author) to convolution, starting from translation invariant linear operators $T$ on $\left(C_0(\mathbb{R}^d), \|\cdot\|_\infty\right)$, which commute with translations. One can argue that these operators model so-called BIBOS systems, because one assumes that (up to some constant) the upper bound of the output signal can be controlled by the upper bound of the input signal. In addition, finite duration input signals $f \in C_c(\mathbb{R}^d)$ are assumed to produce output signals $T(f)$, which decay at infinity.*

## 2. Characterization of TILS as Convolution Operators

We now come to the first main result directly connected with the title of this paper. It provides a characterization of translation invariant operators on $\left(C_0(\mathbb{R}^d), \|\cdot\|_\infty\right)$. For this purpose, let us briefly remind readers of the *flip-operator* given by $f^\vee(z) = f(-z)$ and, correspondingly, $\mu^\vee(f) := \mu(f^\vee)$. This is the natural way to extend the inversion at

the group level to the set of all (first discrete, then general) bounded measures, since one obviously has $\delta_x{}^{\vee} = \delta_{-x}$, $x \in \mathbb{R}^d$.

**Definition 1.** *Given $\mu \in \boldsymbol{M}_b(\mathbb{R}^d)$, we define the convolution operator $C_\mu$ by*

$$C_\mu f(z) = C_\mu(f)(z) := \mu(T_z f^{\vee}) = \mu((T_{-z}f)^{\vee}) = T_z(\mu^{\vee})(f). \tag{5}$$

*Any convolution operator can be viewed as a moving average by setting $\nu := \mu^{\vee}$ :*

$$C_\mu f(z) = [T_z \nu](f), \quad f \in \boldsymbol{C}_0(\mathbb{R}^d). \tag{6}$$

This convention allows us to characterize $\boldsymbol{H}_G(\boldsymbol{C}_0(\mathbb{R}^d))$ (see [4] for details):

**Theorem 3** (Characterization of TILSs on $\boldsymbol{C}_0(\mathbb{R}^d)$)**.** *There is a natural isometric isomorphism between the Banach space $\boldsymbol{H}_G(\boldsymbol{C}_0(\mathbb{R}^d))$, endowed with the operator norm, and $(\boldsymbol{M}_b(\mathbb{R}^d), \|\cdot\|_{\boldsymbol{M}_b})$, the dual of $(\boldsymbol{C}_0(\mathbb{R}^d), \|\cdot\|_\infty)$, by the following pair of mappings:*

1. *Given a bounded measure $\mu \in \boldsymbol{M}_b(\mathbb{R}^d)$, we define the operator $C_\mu$ via (5) above;*
2. *Conversely, we define for $T \in \boldsymbol{H}_G(\boldsymbol{C}_0(\mathbb{R}^d))$ the linear functional $\mu = \mu_T$ by*

$$\mu_T(f) = [T(f^{\vee})](0). \tag{7}$$

*The mappings $C : \mu \mapsto C_\mu$ and $T \mapsto \mu_T$ are linear, non-expansive, and inverse to each other. Consequently, they establish an isometric isomorphism between the two Banach spaces with*

$$\|\mu_T\|_{\boldsymbol{M}_b} = \|T\|_{\boldsymbol{L}(\boldsymbol{C}_0(\mathbb{R}^d))} \quad and \quad \|C_\mu\|_{\boldsymbol{L}(\boldsymbol{C}_0(\mathbb{R}^d))} = \|\mu\|_{\boldsymbol{M}_b}. \tag{8}$$

As pointed out in [4], this identification can be used as a basis for the definition of *convolution* in $\boldsymbol{M}_b(\mathbb{R}^d)$ by transferring the composition law for operators to the measures generating them. The use of the flip-operator within this identification has the advantage that $C_{\delta_x} = T_x$, i.e., convolution by a Dirac measure corresponds to ordinary translation. One can argue that the rule introduced in Definition 1 is the most natural way to extend the basic composition law

$$\delta_x * \delta_y = \delta_{x+y}, \quad x, y \in \mathbb{R}^d$$

to general measures because it is possible to approximate a general convolution operator by finite linear combinations of translation operators, i.e., by a convolution by some finite, discrete measure (see [4]).

The same principle is used then in [13] in order to demonstrate that (not only $\boldsymbol{L}^1(\mathbb{R}^d)$ or more generally $\boldsymbol{L}^1(G)$, but in fact) $(\boldsymbol{M}_b(\mathbb{R}^d), \|\cdot\|_{\boldsymbol{M}_b})$, viewed as a (commutative) Banach algebra with respect to *convolution* acts in a natural way, often called *integrated group action* on so-called *homogeneous Banach spaces*. From an abstract point of view, these are Banach spaces on which a given group $G$ (we restrict our attention to $G = \mathbb{R}^d$ here) acts in an isometric and strongly continuous way. This concept has its roots in the book of Y. Katznelson [16], where they are defined as Banach spaces $(\boldsymbol{B}, \|\cdot\|_{\boldsymbol{B}})$ of locally integrable functions, with a norm that is isometrically translation invariant and with the additional property that $\lim_{x \to 0} \|T_x f - f\|_{\boldsymbol{B}} = 0$ for all $f \in \boldsymbol{B}$.

Just for the convenience of the reader we provide the abstract statement of Theorem 2 of [13] below, but without describing the technical details because we only want to use it for $(\boldsymbol{B}, \|\cdot\|_{\boldsymbol{B}}) = (\boldsymbol{L}^p(G), \|\cdot\|_p)$ or $(\boldsymbol{A}_p(G), \|\cdot\|_{\boldsymbol{A}_p(G)})$. For simplicity, we restrict our attention to the familiar *regular representation* of the additive group $(\mathbb{R}^d, +)$ on $(\boldsymbol{L}^p(\mathbb{R}^d), \|\cdot\|_p)$:

$$\rho(x)(f) = T_x f, \quad x \in \mathbb{R}^d. \tag{9}$$

This is, in fact, a representation of the group (due to the trivial law $T_x \circ T_y = T_{x+y}$, $x, y \in \mathbb{R}^d$). In the notation of [2], Section 1.2, this corresponds to the choice $\rho = \lambda^p_{\mathbb{R}^d}$.

For $1 \leq p < \infty$, the space $C_c(\mathbb{R}^d)$ is dense in $(L^p(\mathbb{R}^d), \|\cdot\|_p)$ and, hence, in this case, translation is strongly continuous, or equivalently

$$\lim_{x \to 0} \|f - T_x f\|_{L^p(\mathbb{R}^d)} = 0, \quad f \in L^p(\mathbb{R}^d). \tag{10}$$

The abstract version of the general principle derived in [13] reads as follows:

**Theorem 4.** *Any abstract homogeneous Banach space $(\boldsymbol{B}, \|\cdot\|_{\boldsymbol{B}})$ with respect to a given strongly continuous and isometric representation $\rho$ of a locally compact group $G$ is also a Banach module over the Banach algebra $(\boldsymbol{M}_b(G), \|\cdot\|_{\boldsymbol{M}_b})$ (with respect to convolution). This claim includes the validity of following associativity law:*

$$\rho(\mu_1 \star \mu_2) = \rho(\mu_1) \circ \rho(\mu_2), \quad \mu_1, \mu_2 \in \boldsymbol{M}_b(G). \tag{11}$$

*The mapping $(\mu, f) \mapsto \mu \bullet_\rho f = \rho(\mu)f$ is the natural extension of the action of discrete measures given by $\delta_x \bullet_\rho f = \rho(x)f$, and satisfies the norm estimate*

$$\|\mu \bullet_\rho f\|_{\boldsymbol{B}} \leq \|\mu\|_{\boldsymbol{M}} \|f\|_{\boldsymbol{B}}, \quad \mu \in \boldsymbol{M}_b(G), f \in \boldsymbol{B}. \tag{12}$$

We are only interested in the following corollary (cf. Scholium 3 (p. 47) of [2]):

**Corollary 1.** *For $1 \leq p < \infty$, we have: $(\boldsymbol{L}^p(\mathbb{R}^d), \|\cdot\|_p)$ is a Banach module over the Banach algebra $(\boldsymbol{M}_b(\mathbb{R}^d), \|\cdot\|_{\boldsymbol{M}_b})$ with respect to convolution. In particular, it is a Banach module over the Banach convolution algebra $(\boldsymbol{L}^1(\mathbb{R}^d), \|\cdot\|_1)$, in fact, an essential one. Since $(\boldsymbol{L}^1(\mathbb{R}^d), \|\cdot\|_1)$ has bounded approximate units (so-called Dirac sequences), we even have $\boldsymbol{L}^1(\mathbb{R}^d) * \boldsymbol{L}^p(\mathbb{R}^d) = \boldsymbol{L}^p(\mathbb{R}^d)$ by the Cohen–Hewitt Factorization Theorem, and also*

$$\lim_{\rho \to 0} \|f - \mathrm{St}_\rho g * f\|_{\boldsymbol{L}^p(\mathbb{R}^d)} = 0, \quad f \in \boldsymbol{L}^p(\mathbb{R}^d),$$

*for any $g \in \boldsymbol{L}^1(\mathbb{R}^d)$ with $\widehat{g}(0) = \int_{\mathbb{R}^d} g(x)dx = 1$.*

The above corollary includes the well-known estimate as a special case:

$$\|g * f\|_{\boldsymbol{L}^p(\mathbb{R}^d)} \leq \|g\|_{\boldsymbol{L}^1(\mathbb{R}^d)} \|f\|_{\boldsymbol{L}^p(\mathbb{R}^d)}, \quad g \in \boldsymbol{L}^1(\mathbb{R}^d), f \in \boldsymbol{L}^p(\mathbb{R}^d). \tag{13}$$

**Remark 8.** *These two extreme cases already indicate that any $T \in \boldsymbol{H}_G(\boldsymbol{L}^1(\mathbb{R}^d))$ extends to a multiplier for $(\boldsymbol{L}^p(\mathbb{R}^d), \|\cdot\|_p)$, in particular for $p = 2$ (with transfer function $h = \widehat{\mu}$), since we have for $1 \leq p < \infty$:*

$$\|\mu * f\|_{\boldsymbol{L}^p(\mathbb{R}^d)} \leq \|\mu\|_{\boldsymbol{M}_b} \|f\|_{\boldsymbol{L}^p(\mathbb{R}^d)}, \quad f \in \boldsymbol{L}^p(\mathbb{R}^d). \tag{14}$$

**Remark 9.** *Once we define $(\boldsymbol{M}_b(\mathbb{R}^d), \|\cdot\|_{\boldsymbol{M}_b})$ as the Banach dual of $(\boldsymbol{C}_0(\mathbb{R}^d), \|\cdot\|_\infty)$, we have to introduce $(\boldsymbol{L}^1(\mathbb{R}^d), \|\cdot\|_1)$ in the context provided by [4]. Given $k \in \boldsymbol{C}_c(\mathbb{R}^d)$, we define the (absolutely continuous) measure using a simple Riemann integral:*

$$\mu_k(f) := \int_{\mathbb{R}^d} f(x)k(x)dx, \quad f \in \boldsymbol{C}_0(\mathbb{R}^d). \tag{15}$$

*Once this is verified, one has*

$$\|\mu_k\|_{\boldsymbol{M}_b(\mathbb{R}^d)} = \|k\|_{\boldsymbol{L}^1(\mathbb{R}^d)} := \int_{\mathbb{R}^d} |k(x)|dx,$$

*It is not difficult to recognize that $(\boldsymbol{L}^1(\mathbb{R}^d), \|\cdot\|_1)$ can be characterized as the norm closure of $\boldsymbol{C}_c(\mathbb{R}^d)$ (with the identification of $k \in \boldsymbol{C}_c(\mathbb{R}^d)$ with $\mu_k \in \boldsymbol{M}_b(\mathbb{R}^d)$) in the Banach space*

$(M_b(\mathbb{R}^d), \|\cdot\|_{M_b})$. *Alternatively, it can be characterized as the set of all bounded measures having continuous shifts, i.e.,*

$$L^1(\mathbb{R}^d) = \{\mu \in M_b(\mathbb{R}^d) \mid \|T_x\mu - \mu\|_{M_b(\mathbb{R}^d)} \to 0 \text{ for } x \to 0\}. \tag{16}$$

*This characterization also implies that it is a closed ideal inside of $M_b(\mathbb{R}^d)$ and, thus, a Banach algebra of its own right, under convolution. Moreover, it has bounded approximate units.*

### 3. Fourier Multipliers between $L^p$-Spaces

The fact that any of the spaces $(L^p(\mathbb{R}^d), \|\cdot\|_p)$ is isometrically invariant under the *reflection operator $f \mapsto f^\vee$* combined with the observation that the adjoint operator of a convolution operator $T(f) = \mu * f$ on $(L^p(\mathbb{R}^d), \|\cdot\|_p)$ is just $T'(h) = \mu^\vee * h$, $h \in (L^q(\mathbb{R}^d), \|\cdot\|_q)$ (the dual space of $(L^p(\mathbb{R}^d), \|\cdot\|_p)$, with $1/p + 1/q = 1$), implies that there is a natural isometric isomorphism between $H_G(L^p(\mathbb{R}^d))$ and $H_G(L^q(\mathbb{R}^d))$ for $1 < p < \infty$ (see [2], Theorem 5 of Chap.1.4). By the method of complex interpolation of Banach spaces (see [17,18]), one can derive the following result:

**Theorem 5.** *Given $p \in (1,2)$ and $T \in H_G(L^p(\mathbb{R}^d))$, then $T$ (restricted e.g., to $L^1(\mathbb{R}^d) \cap L^2(\mathbb{R}^d) \subset L^p(\mathbb{R}^d)$) extends to a bounded linear operator on $(L^r(\mathbb{R}^d), \|\cdot\|_r)$ for any $r \in [p,2]$ (or in fact $r \in [p,q]$), with $1/q + 1/p = 1$ and with uniform control of the norms of all these extensions.*

This result implies that any multiplier for $(L^p(\mathbb{R}^d), \|\cdot\|_p)$ (with $1 < p < \infty$) also defines a multiplier on $(L^2(\mathbb{R}^d), \|\cdot\|_2)$, and thus has a representation as a pointwise multiplier by a uniquely determined $h \in L^\infty(\mathbb{R}^d)$. Finding sufficient conditions for a bounded and continuous function to define a bounded operator on one of the spaces $(L^p(\mathbb{R}^d), \|\cdot\|_p)$ is a delicate question beyond the scope of this note. Taking the distributional description of the situation, one can describe the operator as

$$T(f) = \sigma * f, \quad \text{with } \sigma = \mathcal{F}^{-1}(h). \tag{17}$$

Since $(L^\infty(\mathbb{R}^d), \|\cdot\|_\infty) \hookrightarrow S_0'(\mathbb{R}^d) \hookrightarrow \mathcal{S}'(\mathbb{R}^d)$, we can invoke the inverse Fourier transform, which is an automorphism in the setting of mild respectively tempered distributions; thus, $\mathcal{F}^{-1}(\mathcal{F}L^\infty(\mathbb{R}^d)) = \mathcal{F}(\mathcal{F}L^\infty(\mathbb{R}^d))$ is a well-defined Banach space of tempered distributions with the norm $\|\sigma\|_{\mathcal{F}L^\infty(\mathbb{R}^d)} = \|h\|_\infty$, providing the obvious embedding

$$(M_b(\mathbb{R}^d), \|\cdot\|_{M_b}) \hookrightarrow \left(\mathcal{F}L^\infty(\mathbb{R}^d), \|\cdot\|_{\mathcal{F}L^\infty(\mathbb{R}^d)}\right). \tag{18}$$

The verbal description of this continuous embedding is the plausible statement that any bounded measure is also a pseudomeasure.

Simple cases show that the situation changes drastically if the target space is different from the domain of a multiplier. In such a case, one may have plenty of multipliers that are not represented as pseudomeasures. A typical result reads as follows:

**Lemma 1.** *For $1 < p \le \infty$, we have the following isometric identification:*

$$H_G(L^1, L^p(\mathbb{R}^d)) = L^p(\mathbb{R}^d), \quad \text{via} \quad T(g) = f * g, \; g \in L^1(\mathbb{R}^d), f \in L^p(\mathbb{R}^d). \tag{19}$$

**Remark 10.** *Obviously, any $f \in L^2(\mathbb{R}^d)$ defines a transfer function mapping $\mathcal{F}L^1(\mathbb{R}^d) \subset C_0(\mathbb{R}^d)$ into $L^2(\mathbb{R}^d)$ (in a bounded way), but since there are clearly unbounded functions in $L^2(\mathbb{R}^d)$, it becomes clear that it is not true that multipliers from $L^p(\mathbb{R}^d)$ to $L^q(\mathbb{R}^d)$ (for different values of $p,q$) can be represented as convolutions by pseudomeasures. This lead, during the study of the multiplier problem, to the introduction of the so-called quasimeasures by Garth I. Gaudry (see [19]). We do not repeat the complicated definition here (which was well-motivated by the*

*studies of the multiplier problem in different settings at that time) but recall an important result by M. Cowling [20], who obtained the following characterization: $Q(\mathbb{R}^d)$ can be defined as the dual space of $C_c(\mathbb{R}^d) \cap \mathcal{F}L^1(\mathbb{R}^d)$, endowed with the inductive limit topology. In other words, a linear mapping $\sigma$ from $C_c(\mathbb{R}^d) \cap \mathcal{F}L^1(\mathbb{R}^d)$ defines a quasimeasure if one has the following continuity property: for any given compact set $K \subset \mathbb{R}^d$ and any sequence of compactly supported functions $(k_n)_{n \geq 1}$ with $\mathrm{supp}(k_n) \subset K$ and $\hat{k}_n \in L^1(\mathbb{R}^d)$, with $\lim_{n \to \infty} \|k_n\|_{L^1} = 0$, one has $\sigma(k_n) \to 0$. Obviously one can view $Q(\mathbb{R}^d)$ as a subspace of all distributions on $\mathbb{R}^d$ (in the sense of L. Schwartz), but unfortunately, it is not a subspace of the space $\mathcal{S}'(\mathbb{R}^d)$ of tempered distributions and, thus, a priori quasimeasures may fail to "have a Fourier transform", even in the context of distributions or ultra-distributions.*

Nevertheless, one can summarize the results found in [1] as follows (we are not stating the most general version here):

**Proposition 1.** *Given two parameters $p, q \in (1, \infty)$ and $T \in H_G(L^p, L^q)$, there exist two quasimeasures $\sigma$ and $\tau$, such that one has*

$$T(f) = \sigma * f, \quad f \in C_c(\mathbb{R}^d) \cap \mathcal{F}L^1(\mathbb{R}^d),$$

*and*

$$\mathcal{F}(T(f)) = \tau \cdot \hat{f}, \quad f \in \mathcal{F}^{-1}(\mathcal{D}(\mathbb{R}^d)) \subset \mathcal{S}(\mathbb{R}^d).$$

Note that in the above proposition $p, q$ are not related by duality. Also, the statement does not say anything about the (expected) relationship between $\sigma$ and $\tau$, which would be $\tau = \mathcal{F}(\sigma)$. gfei: This problem arises because, as has been mentioned, a general quasimeasure may not have a Fourier transform, even in the sense of tempered distributions.

However, at least both formulas are meaningful in the following sense. The convolution of $\sigma \in Q(\mathbb{R}^d)$ with a test function $f \in C_c(\mathbb{R}^d) \cap \mathcal{F}L^1(\mathbb{R}^d)$ is well-defined in the pointwise sense, namely, as $\sigma * f(x) = \sigma(T_x f^{\vee}), x \in \mathbb{R}^d$.

Furthermore, $T(f) \in L^q(\mathbb{R}^d) \subset \mathcal{S}'(\mathbb{R}^d)$ has a well-defined Fourier transform in $\mathcal{S}(\mathbb{R}^d) \hookrightarrow \mathcal{D}'(\mathbb{R}^d)$. The pointwise product of $\tau \in Q(\mathbb{R}^d)$ with $\hat{f} \in \mathcal{D}(\mathbb{R}^d) \subset C_c(\mathbb{R}^d) \cap \mathcal{F}L^1(\mathbb{R}^d)$ is also well defined in the usual way, as $\tau \cdot \hat{f}(g) = \tau(\hat{f} \cdot g)$ for $g \in C_c(\mathbb{R}^d) \cap \mathcal{F}L^1(\mathbb{R}^d)$, thanks to the pointwise algebra properties of $(\mathcal{F}L^1(\mathbb{R}^d), \|\cdot\|_{\mathcal{F}L^1})$; hence, $\tau \cdot \hat{f}$ is also a well-defined distribution on $\mathbb{R}^d$.

**Remark 11.** *It is an important observation by Hörmander (see [21]) that there is no non-trivial multiplier from $(L^p(\mathbb{R}^d), \|\cdot\|_p)$ to $(L^q(\mathbb{R}^d), \|\cdot\|_q)$ for $1 \leq q < p < \infty$. Roughly speaking, this fact corresponds to the basic properties of convolution operators. Such operators can be used to increase the smoothness of a given function, but they are not suited to improve the decay. Thus, the simple fact that $(L^p(\mathbb{R}^d), \|\cdot\|_p)$ contains poorly decaying, non-negative functions that do not belong to $(L^q(\mathbb{R}^d), \|\cdot\|_q)$ for $q < p$ indicates that there may be problems. The actual proof uses a somewhat different argument, making use of the global properties of $L^p$-spaces (which are quite different, by the above argument).*

## 4. The Herz–Figa-Talamanca Algebra

Starting from the 60ths, A. Figa-Talamanca and G.I. Gaudry have undertaken detailed studies of the multiplier problem for $L^p(G)$. Their findings have been published in a series of papers [19,22–27]. The key player in this characterization of the space of multipliers (one could also call them convolutors) is a certain space $(A_p(G), \|\cdot\|_{A_p(G)})$, which was been introduced by Figa-Talamanca in [23]. Every linear functional on this Banach space defines a convolution kernel and vice versa. It is also known as *Herz algebra* because C. Herz has shown that it is a Banach algebra with respect to pointwise multiplication [28], but we will not make use of any of these considerations and present a self-contained approach to our main question, namely, the identification of the multiplier space of $(A_p(G), \|\cdot\|_{A_p(G)})$.

Given a LCA group $G$ and $p \in [1, \infty)$, we write $q$ for the dual index, with $1/p + 1/q = 1$.

**Definition 2.** *Given $1 < p < \infty$, the Herz–Figa-Talamanca space, as defined by*

$$A_p(G) = \left\{ f \in C_b(G) \,|\, f = \sum_{n \geq 1} f_n * g_n, f_n \in L^p, g_n \in L^q, \quad with \quad \sum_{n \geq 1} \|f_n\|_p \|g_n\|_q < \infty \right\}.$$

*Any such representation, for suitable sequences $(f_n)_{n \geq 1}$ in $L^p(G)$ and $(g_n)_{n \geq 1}$ in $L^q(G)$, is called an admissible representation of $f \in A_p(G)$. The natural norm is given by*

$$\|f\|_{A_p(G)} = \inf\left\{ \sum_{n \geq 1} \|f_n\|_p \|g_n\|_q \right\}, \tag{20}$$

*where the infimum is taken over all admissible representations.*

From the definition, it is not difficult to verify the following facts (see [2]):

**Lemma 2.** *Let $G$ be a LCA group, and $1 < p < \infty$. Then one has:*

1. *$\left(A_p(G), \|\cdot\|_{A_p(G)}\right)$ is continuously embedded into $\left(C_0(G), \|\cdot\|_\infty\right)$;*
2. *$\left(A_p(G), \|\cdot\|_{A_p(G)}\right)$ is a Banach space;*
3. *The compactly supported functions, in fact, even $\mathcal{F}L^1 \cap C_c(G)$ and hence $S_0(G)$, are a dense subspace of $\left(A_p(G), \|\cdot\|_{A_p(G)}\right)$;*
4. *Translation and modulation act isometrically on $\left(A_p(G), \|\cdot\|_{A_p(G)}\right)$;*
5. *$\lim_{x \to 0} \|T_x f - f\|_{A_p(G)} = 0$ for every $f \in A_p(G)$;*
6. *$\left(A_p(G), \|\cdot\|_{A_p(G)}\right)$ is a homogeneous Banach space, and thus an essential Banach module over $\left(L^1(G), \|\cdot\|_1\right)$ with respect to convolution;*
7. *$\left(A_p(G), \|\cdot\|_{A_p(G)}\right)$ is a pointwise Banach module over $\left(\mathcal{F}L^1(G), \|\cdot\|_{\mathcal{F}L^1(G)}\right)$;*
8. *The reflection operator $f \mapsto f^\vee$ is an isometry on $\left(A_p(G), \|\cdot\|_{A_p(G)}\right)$.*

**Proof.** We only provide the argument for claim (3). Recall that Plancherel's theorem combined with the Cauchy–Schwarz inequality implies that we have, for $f, g \in C_c(G)$, that $\mathcal{F}(f * g) = \hat{f} \cdot \hat{g} \in L^2 \cdot L^2 \subseteq L^1$, or $f * g \in \mathcal{F}L^1 \cap C_c(G)$, but since the local structure of $S_0(G)$ coincides with that of $\mathcal{F}L^1$, we have $C_c(G) * C_c(G) \subset S_0(G)$. □

**Remark 12.** *It is much less obvious that $\left(A_p(G), \|\cdot\|_{A_p(G)}\right)$ is a Banach algebra with respect to pointwise multiplication, the so-called Herz algebra. However, we will not need this fact here. In addition, it is not clear to what extent this property extends to similar constructions, e.g., with $\left(L^p(G), \|\cdot\|_p\right)$ replaced by corresponding Lorentz spaces $L(p,q)(G)$.*

**Remark 13.** *Note that one may assume that the convolution factors $f_n$ and $g_n$ are taken from a dense subspace of $\left(L^p(G), \|\cdot\|_p\right)$ resp. $\left(L^q(G), \|\cdot\|_q\right)$ such as $\mathcal{F}L^1 \cap C_c(G)$, or from the Segal algebra $S_0(G)$, which is a dense subspace of $\left(L^p(G), \|\cdot\|_p\right)$, with corresponding modifications for the norm on $A_p(G)$.*

## 5. The Banach Gelfand Triple

We describe the material for $G = \mathbb{R}^d$ and start from the *Banach Gelfand Triple* $(S_0, L^2, S_0')(\mathbb{R}^d)$, consisting of the Banach algebra $\left(S_0(\mathbb{R}^d), \|\cdot\|_{S_0}\right)$ of test functions, the Hilbert space $\left(L^2(\mathbb{R}^d), \|\cdot\|_2\right)$ and the ambient space $\left(S_0'(\mathbb{R}^d), \|\cdot\|_{S_0'}\right)$ of *mild distributions*, endowed with both the norm and the $w^*$-topology. A point of departure for our note is the simple observation that $L^p$-spaces, their Fourier transforms, and also their Fourier

multipliers, etc., can be well described in this context (see [29]). For any $1 \le p \le \infty$, one has

$$\left(\boldsymbol{S}_0(\mathbb{R}^d), \|\cdot\|_{\boldsymbol{S}_0}\right) \hookrightarrow \left(\boldsymbol{L}^p(\mathbb{R}^d), \|\cdot\|_p\right) \hookrightarrow \left(\boldsymbol{S}_0'(\mathbb{R}^d), \|\cdot\|_{\boldsymbol{S}_0'}\right). \tag{21}$$

One of the basic results of our study is the observation (already formulated as Theorem C2 in [30] and then published, among others, as Theorem 8.4 in [9]), showing that any multiplier from $\boldsymbol{S}_0(\mathbb{R}^d)$ to $\boldsymbol{S}_0'(\mathbb{R}^d)$ is a convolution operator by some $\sigma \in \boldsymbol{S}_0'(\mathbb{R}^d)$:

$$T(f)(x) = T_x \sigma^{\checkmark}(f) = (T_{-x}\sigma)^{\checkmark}(f) = T_{-x}\sigma(f^{\checkmark}) = \sigma(T_x f^{\checkmark}) \quad \forall f \in \boldsymbol{S}_0(\mathbb{R}^d), x \in \mathbb{R}^d, \tag{22}$$

In compact form, it is given in the pointwise sense by

$$Tf(x) = \sigma(T_x(f^{\checkmark})), \quad x \in \mathbb{R}^d, f \in \boldsymbol{S}_0(\mathbb{R}^d). \tag{23}$$

Combining these two observations, one easily verifies that any bounded linear operator on any of the spaces $\left(\boldsymbol{L}^p(\mathbb{R}^d), \|\cdot\|_p\right)$ with $1 \le p < \infty$, can be realized as a convolution operator (in the sense of (23)) for a uniquely determined $\sigma \in \boldsymbol{S}_0'(\mathbb{R}^d)$. In engineering terminology, the "convolution kernel" (or distribution) $\sigma$ is called the *impulse response* of the linear, translation invariant system $T$. This is most often the case for $p = 2$, or $p = 1$, or equivalently, BIBOS (Bounded Input giving Bounded Output systems) systems, with $\left(\boldsymbol{L}^\infty(\mathbb{R}^d), \|\cdot\|_\infty\right)$ replaced by $\left(\boldsymbol{C}_0(\mathbb{R}^d), \|\cdot\|_\infty\right)$, see [4]. Combined with the Fourier transform, which is well-defined on $\left(\boldsymbol{S}_0'(\mathbb{R}^d), \|\cdot\|_{\boldsymbol{S}_0'}\right)$ via the usual rule (based on the Fourier invariance of $\left(\boldsymbol{S}_0(\mathbb{R}^d), \|\cdot\|_{\boldsymbol{S}_0}\right)$!)

$$\widehat{\sigma}(f) = \sigma(\widehat{f}), \quad f \in \boldsymbol{S}_0(\mathbb{R}^d), \tag{24}$$

It is clear that these operators can also be viewed as Fourier multipliers with $h = \widehat{\sigma}$ as *transfer function* of the operator/system $T$. Since (24) defines the extended Fourier as the transpose of the classical Fourier transform, which leaves $\boldsymbol{S}_0(\mathbb{R}^d)$ invariant, it is a $w^*$-$w^*$-continuous operator on $\boldsymbol{S}_0'(\mathbb{R}^d)$. The $w^*$−density of $\boldsymbol{S}_0(\mathbb{R}^d)$ in $\boldsymbol{S}_0'(\mathbb{R}^d)$ then implies that it the unique extension of this classical Fourier transform with this property.

The elements of $\boldsymbol{S}_0'(\mathbb{R}^d)$, the so-called *mild distributions*, are members of the dual of a decent *Segal algebra* (see [31–33]), whose local structure is that of the Fourier Algebra $\left(\mathcal{F}\boldsymbol{L}^1(\mathbb{R}^d), \|\cdot\|_{\mathcal{F}\boldsymbol{L}^1}\right)$. Thus, it is clear how to define the support $\operatorname{supp}(\sigma)$, and, consequently, the *spectrum* of $\sigma \in \boldsymbol{S}_0'$: simply as $\operatorname{supp}(\widehat{\sigma})$.

The following theorem is key for the main results of this manuscript for LCA groups. For simplicity, we only reproduce a direct proof for the case $G = \mathbb{R}^d$.

**Theorem 6.** *For any LCA group $G$, there is a natural isomorphism between $\boldsymbol{H}_G(\boldsymbol{S}_0, \boldsymbol{S}_0')(G)$ and $\boldsymbol{S}_0'(G)$ given by the following linear isometries, which are inverse to each other (here the symbol $T_x f^{\checkmark}$ stands for $T_x(f^{\checkmark})$):*

$$\sigma \mapsto C_\sigma : \quad C_\sigma(f)(x) = \sigma(T_x f^{\checkmark}), \quad x \in G, \tag{25}$$

*and*

$$T \mapsto \sigma_T : \quad \sigma_T(f) = [Tf^{\checkmark}](0), \quad f \in \boldsymbol{S}_0(G). \tag{26}$$

*Moreover, the ultra-weak convergence (this concept will be explained below, see Definition 4) of a (bounded) net of operators $C_{\sigma_\alpha}$ corresponds in a one-to-one way to the $w^*$-convergence of the corresponding distributional kernels $(\sigma_\alpha)$ in $\boldsymbol{S}_0'(G)$, which generate these convolution operators. In compact form, this claim can be summarized as*

$$\boldsymbol{H}_G(\boldsymbol{S}_0, \boldsymbol{S}_0') = \boldsymbol{H}_G(\boldsymbol{S}_0, \boldsymbol{C}_b) \quad and \quad \|\|T\|\|_{\boldsymbol{S}_0 \to \boldsymbol{S}_0'} \approx \|\sigma\|_{\boldsymbol{S}_0'}. \tag{27}$$

**Proof of Theorem 6.** Due to the Fourier invariance of $\boldsymbol{S}_0(\mathbb{R}^d)$, we have: $f \in \boldsymbol{S}_0(\mathbb{R}^d)$ if and only if $\widehat{f} \in \boldsymbol{S}_0(\mathbb{R}^d) = \boldsymbol{W}(\mathcal{F}\boldsymbol{L}^1, \ell^1)(\mathbb{R}^d)$. This means that for some (any) function

$\psi \in \mathcal{F}L^1 \cap C_c(\mathbb{R}^d)$ that generates a so-called BUPU, i.e., which satisfies $\sum_{k \in \mathbb{Z}^d} \psi(x-k) \equiv 1$, $\widehat{f}$ can be written as an absolutely convergent series in $(\mathcal{F}L^1(\mathbb{R}^d), \|\cdot\|_{\mathcal{F}L^1})$ of the form

$$\widehat{f} = \sum_{k \in \mathbb{Z}^d} (T_k \psi) \cdot \widehat{f},$$

However, since functions in $\mathcal{F}L^1(\mathbb{R}^d)$ define pointwise multipliers of $W(\mathcal{F}L^1, \ell^1)(\mathbb{R}^d)$, it is easy to show that the sum is actually absolutely convergent in $(S_0(\mathbb{R}^d), \|\cdot\|_{S_0})$. Choosing any $\psi^* \in S_0(\mathbb{R}^d)$ (e.g., $\psi^* \in \mathcal{S}(\mathbb{R}^d)$) with $\psi(s) \cdot \psi^*(s) = \psi(s)$ for all $s \in \mathbb{R}^d$, one finds that

$$\|T_k \psi \cdot \widehat{f}\|_{S_0(\mathbb{R}^d)} = \|T_k \psi^* \cdot T_k \psi \cdot \widehat{f}\|_{S_0(\mathbb{R}^d)} \leq \|T_k \psi^*\|_{S_0} \|T_k \psi \cdot \widehat{f}\|_{\mathcal{F}L^1}$$

and, consequently, one has for any $f \in S_0(\mathbb{R}^d)$:

$$\sum_{k \in \mathbb{Z}^d} \|(T_k \psi) \cdot \widehat{f}\|_{S_0} \leq \|\psi^*\|_{S_0} \sum_{k \in \mathbb{Z}^d} \|(T_k \psi) \cdot \widehat{f}\|_{\mathcal{F}L^1} \leq C \|\psi^*\|_{S_0} \|f\|_{S_0}.$$

Note that $\varphi^* := \mathcal{F}^{-1} \psi^*$ also belongs to $S_0(\mathbb{R}^d)$ and that we have now

$$f = \sum_{k \in \mathbb{Z}^d} M_k \varphi^* * M_k \varphi * f.$$

Given $T \in L(S_0, S_0')$, a bounded linear operator that commutes with translation and hence with convolution by $M_k \varphi^*$ for each $k \in \mathbb{Z}^d$, we obtain:

$$T(f) = \sum_{k \in \mathbb{Z}^d} M_k \varphi^* * T[M_k \varphi * f], \tag{28}$$

and further

$$\|T(f)\|_\infty \leq \|\varphi^*\|_{S_0} \|T\|_{S_0 \to S_0'} \sum_{k \in \mathbb{Z}^d} \sum_{n \in \mathbb{Z}^d} \|M_k \varphi * f\|_{S_0}, \tag{29}$$

or altogether; for some constant $C > 0$, one has

$$\|T(f)\|_\infty \leq C \|T\|_{S_0 \to S_0'} \|f\|_{S_0}, \quad \forall f \in S_0. \tag{30}$$

Since we have $\|T_x f - f\|_{S_0(\mathbb{R}^d)} \to 0$ for $x \to 0$ for any $f \in S_0(\mathbb{R}^d)$, this also implies the *uniform continuity* of $Tf$ for each $f \in S_0(\mathbb{R}^d)$ by the following argument:

$$\|T_x(Tf) - Tf\|_\infty = \|T(T_x f - f)\|_\infty \to 0, \quad \text{as } x \to 0.$$

The rest of the argument now follows the usual procedure. We just have to check that $\sigma(f) := T(f^{\vee})(0)$ is a well-defined linear function on $(S_0(\mathbb{R}^d), \|\cdot\|_{S_0})$ and, in fact, can be used to *represent* the linear operator. We leave the details to the interested reader. □

As an immediate corollary to this result, we have the following consequences for multipliers between (potentially different) $L^p$-spaces over $\mathbb{R}^d$.

**Theorem 7.** *Given $p \in [1, \infty)$ and $r \in [1, \infty]$, any bounded linear operator $T$ from $(L^p(\mathbb{R}^d), \|\cdot\|_p)$ to $(L^r(\mathbb{R}^d), \|\cdot\|_r)$ that commutes with all translations $T_y, y \in \mathbb{R}^d$ is a convolution operator by a uniquely determined $\sigma \in S_0'(\mathbb{R}^d)$, meaning that*

$$Tf(x) = \sigma * f(x) = \sigma(T_x f^{\vee}), \; f \in S_0(\mathbb{R}^d), x \in \mathbb{R}^d. \tag{31}$$

*Equivalently, $T$ can be represented as Fourier multiplier: $\widehat{Tf} = \tau \cdot \widehat{f}, f \in S_0(\mathbb{R}^d)$, for $\tau = \widehat{\sigma} \in S_0'(\mathbb{R}^d)$ and vice versa (with $\sigma = \mathcal{F}^{-1}(\tau)$).*

Note that we do not claim that (31) is valid for all $f \in L^p(\mathbb{R}^d)$. Of course, it is enough to explicitly describe the action of a bounded linear operator on a dense subspace, while the extension to the full Banach space is carried out by approximation ($S_0(\mathbb{R}^d)$ is dense in $(L^p(\mathbb{R}^d), \|\cdot\|_p)$ for $p < \infty$).

An important example for a non-trivial Fourier multiplier is the chirp signal $ch(t) = exp(i\pi x^2) \in C_b(\mathbb{R})$. It has the interesting property of being mapped onto itself by the (generalized) Fourier transform. Although $SINC := \mathcal{F}^{-1}(box)$ evidently belongs to $L^2(\mathbb{R})$ but not to $L^1(\mathbb{R})$, it is *not possible* to describe the action of $\sigma = ch$ via convolution on the SINC function by a pointwise integral, because $\int_{\mathbb{R}} ch(t) T_x(SINC)(t) dt$ does not exist (in the Lebesgue sense) for any $x \in \mathbb{R}$.

We can take the definition of $A_p(G)$ or $A_p(\mathbb{R}^d)$ as usual (a so-called *convolution tensor product*) and find immediately that this is a Banach space with the usual quotient norm (infimum over all possible representations). In fact, one could use any *homogeneous Banach space* in the spirit of Y. Katznelson, ref. [16] (or the abstract version by H.S. Shapiro; see [34]).

## 6. The Multipliers of the Herz Algebra

The following elementary lemma plays a crucial role in our approach. It might be given as a good exercise to students, but due to its role for the approach presented and for the convenience of the readers, we include a detailed argument. It also provides the basis of future extensions. Let us mention immediately that an important feature is the fact that the target space consists of continuous functions; hence, point evaluations make sense. In this way, one of the key steps in the proof of [4], namely, the identification of the convolution kernel $\tau$ via the simple choice $\tau(f) = T(f^{\vee})(0)$, makes sense.

**Lemma 3.** *For $1 < p < \infty$ and $1/p + 1/q = 1$, one has the following isometric identifications:*

$$H_{\mathbb{R}^d}(L^p(\mathbb{R}^d), C_0(\mathbb{R}^d)) \equiv L^q(\mathbb{R}^d), \tag{32}$$

*and*

$$H_{\mathbb{R}^d}(A_p(\mathbb{R}^d), C_b(\mathbb{R}^d)) \equiv A_p(\mathbb{R}^d)' =: PM_p(\mathbb{R}^d). \tag{33}$$

**Proof.** We have to verify two inclusions. The first inclusion is the obvious one, namely, the claim that functions $g \in L^q(\mathbb{R}^d)$ define multipliers into $C_0(\mathbb{R}^d)$. For this, we observe that for $f \in L^p(\mathbb{R}^d)$ and $g \in L^q(\mathbb{R}^d)$

$$L^p(\mathbb{R}^d) * L^q(\mathbb{R}^d) \subset C_0(\mathbb{R}^d), \quad \text{and} \quad \|f * g\|_\infty \le \|f\|_{L^p(\mathbb{R}^d)} \|g\|_{L^q(\mathbb{R}^d)}. \tag{34}$$

This results from the following easy observation. First of all, Hölder's inequality provides us with the (pointwise) estimation

$$\|f * g\|_\infty = \sup_{x \in \mathbb{R}^d} |f * g(x)| \le \|f\|_{L^p(\mathbb{R}^d)} \|g\|_{L^q(\mathbb{R}^d)}, \quad x \in \mathbb{R}^d, \tag{35}$$

and the fact that $T_x(f * g) = T_x f * g$, $x \in \mathbb{R}^d$, which implies, in conjunction with (35), that

$$\|T_x(f * g) - f * g\|_\infty \le \|T_x f - f\|_{L^p} \|g\|_{L^q} \to 0 \text{ for } x \to 0. \tag{36}$$

The density of $C_c(\mathbb{R}^d)$ in both $(L^p(\mathbb{R}^d), \|\cdot\|_p)$ and $(L^q(\mathbb{R}^d), \|\cdot\|_q)$ justifies, on the one hand, that $\|T_x f - f\|_{L^p(\mathbb{R}^d)}$ for $x \to 0$, thus showing that $f * g$ is in fact uniformly continuous, i.e., belongs to $C_{ub}(\mathbb{R}^d)$. On the other hand, it implies, again using the norm estimate (34), that functions in $C_c(\mathbb{R}^d) * C_c(\mathbb{R}^d) \subset C_c(\mathbb{R}^d) \subset C_0(\mathbb{R}^d)$ can be used to approximate $f * g$ in $(C_b(\mathbb{R}^d), \|\cdot\|_\infty)$ in the given situation. Since $(C_0(\mathbb{R}^d), \|\cdot\|_\infty)$ is the closure of $C_c(\mathbb{R}^d)$ in $(C_b(\mathbb{R}^d), \|\cdot\|_\infty)$, the first inclusion is realized.

Let us now address the converse relation. Given $T \in H_{\mathbb{R}^d}(L^p(\mathbb{R}^d), C_0(\mathbb{R}^d)) \subset H_{\mathbb{R}^d}(S_0(\mathbb{R}^d), S_0'(\mathbb{R}^d))$, we know that the linear functional $\tau$ given by $\tau(f) = T(f^{\vee})(0)$ defines the convolu-

tion operator $T$, at least on $S_0(\mathbb{R}^d) \subset L^p(\mathbb{R}^d)$ via convolution, or $Tf(x) = \tau(T_x f^{\vee}), x \in \mathbb{R}^d$. Since $S_0(\mathbb{R}^d)$ is dense in $(L^p(\mathbb{R}^d), \|\cdot\|_p)$, we can use the pointwise estimate

$$|\tau(f)| \leq \|T(f^{\vee})\|_{\infty} \leq \||T\||_{L^p(\mathbb{R}^d) \to C_0(\mathbb{R}^d)} \|f\|_{L^p(\mathbb{R}^d)},$$

to verify that $\tau$ is, in fact, a regular distribution realized by some function $g \in L^q(\mathbb{R}^d)$ according to the standard duality result $((L^p(\mathbb{R}^d), \|\cdot\|_p))' \equiv (L^q(\mathbb{R}^d), \|\cdot\|_q)$.

This concludes the proof of the first statement. The second one follows along similar lines. □

We now come to the main result of this paper. Since all the ingredients needed are available in the setting of LCA groups, we formulate it in the context of LCA groups.

**Theorem 8.** *For any LCA group $G$ and $1 < p < \infty$, one has equality of spaces with a natural isometry of the corresponding (operator) norms:*

$$\boldsymbol{H}_G(\boldsymbol{L}^p(G)) = \boldsymbol{H}_G(\boldsymbol{A}_p(G)) = \boldsymbol{H}_G(\boldsymbol{A}_p(G), \boldsymbol{C}_0(G)) = \boldsymbol{H}_G(\boldsymbol{A}_p(G), \boldsymbol{C}_b(G)). \qquad (37)$$

**Proof.** The proof will proceed by providing a chain of inclusions between the spaces listed in (37), finishing at the end with the inclusion $\boldsymbol{H}_G(\boldsymbol{A}_p(G), \boldsymbol{C}_b(G)) \hookrightarrow \boldsymbol{H}_G(\boldsymbol{L}^p(G))$.

Since both $(\boldsymbol{A}_p(G), \|\cdot\|_{\boldsymbol{A}_p(G)})$ and $(\boldsymbol{L}^p(G), \|\cdot\|_p)$ contain $S_0(G)$ as a dense subspace and all the involved spaces (including $(\boldsymbol{C}_b(G), \|\cdot\|_{\infty})$) are continuously embedded into $(S_0'(G), \|\cdot\|_{S_0'})$, one can treat any multiplier space in the chain as a subspace of $\boldsymbol{H}_G(S_0, S_0')$ resp. $S_0'(G)$, via the identification described in Theorem 6. The main task is therefore to show that continuity of such a convolution operator in one way (defined on the common domain $S_0(\mathbb{R}^d)$) implies continuity in the subsequent one (and the last one implies the continuity in the sense of $(\boldsymbol{L}^p(\mathbb{R}^d), \|\cdot\|_p)$). Recall that for any $\sigma \in S_0'(\mathbb{R}^d)$, the convolution product is given by $\sigma * f(x) = \sigma(T_x f^{\vee})$ for $f \in S_0(\mathbb{R}^d)$, and thus generates a bounded and (uniformly) continuous function on $\mathbb{R}^d$.

(a) Let us first assume that $T$ is defined on $S_0(G)$ and continuous with respect to the $\boldsymbol{L}^p(G)$-norm. Given $\varepsilon > 0$ and $h \in \boldsymbol{A}_p(G)$, we find an admissible representation of $h$

$$h = \sum_{n \geq 1} f_n * g_n \quad \text{with} \quad \sum_{n \geq 1} \|f_n\|_p \|g_n\|_q < \|h\|_{\boldsymbol{A}_p(G)} + \varepsilon.$$

Since $T(f) = \sigma * f$ and the series is absolutely convergent in $(\boldsymbol{A}_p(G), \|\cdot\|_{\boldsymbol{A}_p(G)})$, we have

$$T(h) = \sum_{n \geq 1} T(f_n * g_n) = \sum_{n \geq 1} \sigma * (f_n * g_n) = \sum_{n \geq 1} (\sigma * f_n) * g_n,$$

using the associativity of convolution. Consequently, $T(h) \in \boldsymbol{A}_p(G)$ with

$$\|T(h)\|_{\boldsymbol{A}_p(G)} \leq \sum_{n \geq 1} \|\sigma * f_n\|_p \|g_n\|_q \leq \||T\||_p (1 + \varepsilon) \|h\|_{\boldsymbol{A}_p(G)}.$$

In other words, $T$ is also bounded on $(\boldsymbol{A}_p(G), \|\cdot\|_{\boldsymbol{A}_p(G)})$ and

$$\||T\||_{\boldsymbol{A}_p(G)} \leq \||T\||_{\boldsymbol{L}^p(G)}. \qquad (38)$$

(b) The inclusion $\boldsymbol{H}_G(\boldsymbol{A}_p(G)) \subseteq \boldsymbol{H}_G(\boldsymbol{A}_p(G), \boldsymbol{C}_0(G)) \subseteq \boldsymbol{H}_G(\boldsymbol{A}_p(G), \boldsymbol{C}_b(G))$ follows from the continuous embedding $(\boldsymbol{A}_p(G), \|\cdot\|_{\boldsymbol{A}_p(G)}) \hookrightarrow (\boldsymbol{C}_0(G), \|\cdot\|_{\infty}) \hookrightarrow (\boldsymbol{C}_b(G), \|\cdot\|_{\infty})$.

(c) Let us assume now, conversely, that $T : f \mapsto \sigma * f, f \in S_0(G)$, defines a bounded convolution operator from $(\boldsymbol{A}_p(G), \|\cdot\|_{\boldsymbol{A}_p(G)})$ to $(\boldsymbol{C}_b(G), \|\cdot\|_{\infty})$.

Using the fact that $(\boldsymbol{A}_p(G), \|\cdot\|_{\boldsymbol{A}_p(G)})$ is isometrically invariant under the flip operator, due to the elementary equation

$$(f * g)^{\vee} = f^{\vee} * g^{\vee}, \qquad (39)$$

implying

$$\|h^{\vee}\|_{A_p(G)} = \|h\|_{A_p(G)}, \quad h \in A_p(G), \tag{40}$$

we have

$$|\sigma(h)| = \left| [T(h^{\vee})](0) \right| \leq \|T(h^{\vee})\|_{\infty} \leq \|\|T\|\|_{A_p(G) \to C_b(G)} \|h\|_{A_p(G)}, \tag{41}$$

which, in turn, gives us

$$\|\sigma\|_{A_p'(G)} \leq \|\|T\|\|_{A_p(G) \to C_b(G)}. \tag{42}$$

(d) Finally let us assume that $C_\sigma : f \mapsto \sigma * f$ defines a bounded convolution operator from $S_0(G)$, endowed with the norm of $(A_p(G), \|\cdot\|_{A_p(G)})$) to $(C_b(G), \|\cdot\|_{\infty})$ (or equivalently, according to Lemma 3 that we have $\sigma \in A_p'(G)$). Our goal is to show that $C_\sigma$ defines a bounded linear operator on $(L^p(G), \|\cdot\|_p)$, which obviously also commutes with translations.

Applying the estimate (41) to $h = f * g$, with $f, g \in S_0(G)$ and using the identity

$$\sigma * (f * g) = (\sigma * f) * g \in C_b(G) \tag{43}$$

we come up with the following estimate:

$$\|(\sigma * f) * g\|_{\infty} \leq \|\sigma\|_{A_p'(G)} \|f * g\|_{A_p(G)} \leq \|\sigma\|_{A_p'(G)} \|f\|_p \|g\|_q, \quad f, g \in S_0(G). \tag{44}$$

Since $1 < q < \infty$ Lemma 3 implies that $\sigma * f \in L^p(\mathbb{R}^d)$, with the estimate

$$\|\sigma * f\|_p \leq \|\sigma\|_{A_p'(G)} \|f\|_p, \quad f \in S_0(G), \tag{45}$$

or expressed in terms of the operator norm

$$\|\|C_\sigma\|\|_{L^p(G)} \leq \|\|C_\sigma\|\|_{A_p(G) \to C_b(G)} = \|\sigma\|_{A_p'(G)}. \tag{46}$$

Combined with the estimate (38) we arrive at the following chain of isometries (of course by identifying operators with the joint dense domain $L^p(G) \cap A_p(G)$):

$$\|\|C_\sigma\|\|_{L^p(G)} = \|\|C_\sigma\|\|_{A_p(G)} = \|\sigma\|_{A_p'(G)}. \tag{47}$$

□

The following corollary shows that the above theorem immediately implies an identification of $H_G(L^p(G))$ with the dual of $(A_p(G), \|\cdot\|_{A_p(G)})$ (in [2], the symbol $CV_p(G)$ is used):

**Corollary 2.** *There is an natural isometric identification of operators $T \in H_G(L^p(G))$ with the corresponding convolution kernel $\sigma \in A_p'(G)$, i.e.,*

$$H_G(A_p(G)) = H_G(A_p(G), C_b(G)) \equiv A_p'(G). \tag{48}$$

**Proof.** We only need to point to item (c) in the above proof. □

Combining our results, we have the following key result concerning the Herz algebras:

**Corollary 3.** *There is an isometric isomorphism between the space $CV_p(G) = H_G(L^p(G))$ of convolutors of $(L^p(G), \|\cdot\|_p)$ and $PM_p(G)$, the dual space of $(A_p(G), \|\cdot\|_{A_p(G)})$. Under the (extended) Fourier transform, this space can be identified with the space of p-Fourier multipliers, i.e., the pointwise multipliers on $(\mathcal{F}L^p(G), \|\cdot\|_p)$.*

The pointwise module structure of $A_p(G)$ can be used by duality to transfer it to the dual space. In the context of convolution operators, we can describe it as follows (for simplicity, we choose $G = \mathbb{R}^d$ for the illustration of the principle).

**Corollary 4.** *For any $\varphi \in \mathcal{F}L^1(\mathbb{R}^d)$, we have for any $\sigma \in A_p(\mathbb{R}^d)'$:*

$$\|(\varphi \cdot \sigma) * f\|_{L^p} \leq \|\varphi\|_{\mathcal{F}L^1} \|\sigma\|_{A_p'} \|f\|_{L^p}, \tag{49}$$

*or, in other words,*

$$\|\varphi \cdot \sigma\|_{A_p'} \leq \|\varphi\|_{\mathcal{F}L^1} \|\sigma\|_{A_p'}. \tag{50}$$

**Proof.** We have stated in Lemma 2 (7.) that $A_p$ is a pointwise module over the Fourier algebra $\left(\mathcal{F}L^1(\mathbb{R}^d), \|\cdot\|_{\mathcal{F}L^1}\right)$. Since the adjoint operator on the dual space is just the natural variant of a multiplication operator on such a space, the estimates follow from the corresponding natural estimates for $\left(A_p'(\mathbb{R}^d), \|\cdot\|_{A_p'(\mathbb{R}^d)}\right)$. Technically, they can be derived from the main result of [13] using the fact that $s \mapsto M_s$ is a strongly continuous representation of $\mathbb{R}^d$ (respectively the dual group) on the Banach spaces $\left(A_p(\mathbb{R}^d), \|\cdot\|_{A_p(\mathbb{R}^d)}\right)$. $\square$

For the last result of this paper, we need slightly more, namely, the fact that the usual approximate units for $\left(\mathcal{F}L^1(\mathbb{R}^d), \|\cdot\|_{\mathcal{F}L^1}\right)$ (obtained by dilation) act accordingly in this context.

**Lemma 4.** *Given any $\tau \in S_0(\mathbb{R}^d)$ with $\tau(0) = 1$, and $\sigma \in A_p'(\mathbb{R}^d)$, we have for any $f \in L^p(\mathbb{R}^d)$:*

$$\|(D_\rho \tau \cdot \sigma) * f - \sigma * f\|_{L^p} \to 0, \text{ for } \rho \to 0, \quad \forall f \in L^p(\mathbb{R}^d). \tag{51}$$

**Proof.** Since the dilation operator $D_\rho$ is isometric on $\left(\mathcal{F}L^1(\mathbb{R}^d), \|\cdot\|_{\mathcal{F}L^1}\right)$, the family of convolution operators $(D_\rho \tau \cdot \sigma), \rho \in (0, 1]$ is uniformly bounded on $\left(L^p(\mathbb{R}^d), \|\cdot\|_p\right)$. Hence, we may assume that $\mathcal{F}(\tau)$ is compactly supported and $\mathcal{F}(D_\rho \tau) = \mathrm{St}_\rho(\widehat{\tau})$ has small support.

The fact that the Fourier inversion theorem allows to write

$$\tau(t) = \int_{\mathbb{R}^d} \widehat{\tau}(s) e^{2\pi i t s} ds$$

as an absolutely convergent Riemann integral allows to identify $(\tau \cdot \sigma) * f$ as a vector-valued integral in $\left(L^p(\mathbb{R}^d), \|\cdot\|_p\right)$:

$$(\tau \cdot \sigma) * f = \int_{\mathbb{R}^d} \widehat{\tau}(s)(M_s \sigma * f) ds. \tag{52}$$

Leaving out some (easy) technical details, we can argue that for $f \in L^p(\mathbb{R}^d)$

$$s \mapsto (M_s \sigma) * f = M_s(\sigma * M_{-s} f)$$

is a continuous mapping on $\mathbb{R}^d$ with values in $\left(L^p(\mathbb{R}^d), \|\cdot\|_p\right)$ and, thus, (51) is valid. $\square$

**Remark 14.** *Since the family of convolution operators $f \mapsto (D_\rho \tau \cdot \sigma) * f$, with $\rho \in (0, 1]$, is uniformly bounded and pointwise convergent on $\left(L^p(\mathbb{R}^d), \|\cdot\|_p\right)$, it follows that (51) is not only valid for individual functions $f \in L^p(\mathbb{R}^d)$, but holds uniformly for relatively compact subsets of $\left(L^p(\mathbb{R}^d), \|\cdot\|_p\right)$.*

Finally let us briefly mention the connections to *quasimeasures*. Originally introduced by G. Gaudry in a complicated fashion, Cowling was able to show (see [20]) that they are just locally pseudomeasures; in symbols $Q(G) = \mathcal{F}L_{loc}^\infty(G)$ one has the obvious chain of inclusions: $A_p'(G) \subset S_0'(G) \subset Q(G)$. However, the Fourier image of $A_p'(G)$ under the

Fourier isomorphism from $S_0'(G)$ onto $S_0'(\widehat{G})$ (i.e., the space of Fourier multipliers) also ensures that $\mathcal{F}(A_p'(G)) \subset S_0'(\widehat{G}) \subset Q(\widehat{G})$.

In other words, we have (using engineering terminology) two statements that can be found in the book of Larsen [1] describing the embedding of $H_G(L^p, L^q)$ for $1 < p < q < \infty$ into $Q(\widehat{G})$). The need to go beyond the space of pseudomeasures is expressed by Corollary 5.

**Corollary 5.** *Given $1 \leq p < \infty$, one has: Any multiplier of $L^p(G)$ can be described as a convolution operator by some quasimeasure $\sigma$, but also as a pointwise multiplier of the form $f \mapsto \mathcal{F}^{-1}(\tau \cdot \widehat{f})$ for a suitable quasimeasure $\tau$ on $Q(\widehat{G})$.*

### 7. The Link to p-Pseudomeasures

Inspired by [2], which distinguishes between $CV_p(G)$ and the $PM_p(G)$ (which is first identified with the dual space to $(A_p(G), \|\cdot\|_{A_p(G)})$), while trying to avoid the technical details, we present a result that is closely related to Theorem 6 in Section 4.1 of [2]. For the formulation of this result, we need the following terminology:

**Definition 3.** *Given $1 < p < \infty$, a bounded net $(S_\alpha)_{\alpha \in I}$ of operators in $L(L^p(G))$ is ultra-weakly convergent to some operator $T \in L(L^p(G))$ if one has*

$$\lim_{\alpha \to \infty} \langle S_\alpha f, h \rangle \to \langle Tf, h \rangle, \quad \forall f \in L^p(G), h \in L^q(G).$$

The following definition is also taken from [2]:

**Definition 4.** *Let $G$ be a locally compact group and $1 < p < \infty$. The closure of $\lambda_G^p(M^1(G))$ in $L(L^p(G))$ with respect to the ultra-weak topology is denoted by $PM_p(G)$.*

The following result justifies the identification of $PM_p(G)$ with the dual of space $A_p'(G)$.

**Theorem 9.** *For any $p \in [1, \infty)$, the convolution operators by elements of $S_0(\mathbb{R}^d)$ form an ultra-weak dense subspace of $CV_p(\mathbb{R}^d) = H_G(L^p(\mathbb{R}^d))$.*

**Proof.** Let $f, h$ be given as in Definition 3 and $\varepsilon > 0$. For a given $T \in CV_p(\mathbb{R}^d)$, we know that $Tf = \sigma * f$ for $f \in S_0(\mathbb{R}^d)$, for a suitable distribution $\sigma \in A_p'(\mathbb{R}^d) \subset S_0'(\mathbb{R}^d)$. We can expect that regularized versions of $\sigma$ will do the job, i.e., $f \mapsto h_\rho * f$, for $h_\rho$ of the form $h_\rho = \mathsf{St}_\rho g * (\mathsf{D}_\rho g \cdot \sigma)$ provide such an ultra-weak approximation, where one can choose $g$ to be the Gauss function (or any $g \in S_0(\mathbb{R}^d)$ with $\widehat{g}(0) = 1$, thus more or less with any classical summability kernel; see [35]). Without loss of generality, we may assume $g^\vee = g$ and $g(x) \geq 0$; hence $\|g\|_1 = 1$.

First, we note that the convolution relations for Wiener amalgam spaces in [36] imply

$$h_\rho \in S_0(\mathbb{R}^d) * (S_0(\mathbb{R}^d) \cdot S_0'(\mathbb{R}^d)) \subset W(\mathcal{F}L^1, \ell^1) * W(\mathcal{F}L^\infty, \ell^1) \subset W(\mathcal{F}L^1, \ell^1) = S_0(\mathbb{R}^d).$$

The family of operators $T_\rho : f \mapsto f * h_\rho$ is uniformly bounded on $(L^p(\mathbb{R}^d), \|\cdot\|_p)$ because we have for $f \in S_0(\mathbb{R}^d)$ using $\|\mathsf{D}_\rho g\|_{\mathcal{F}L^1} = \|\mathsf{St}_\rho \widehat{g}\|_1 = \|\widehat{g}\|_1 = \|g\|_1$ and using (50):

$$\|h_\rho * f\|_p = \|[(\mathsf{D}_\rho g \cdot \sigma) * \mathsf{St}_\rho g] * f\|_p = \|[\mathsf{D}_\rho g \cdot \sigma] * (\mathsf{St}_\rho g * f)\|_p \leq \|\mathsf{D}_\rho g \cdot \sigma\|_{A_p'} \|f * \mathsf{St}_\rho g\|_p \tag{53}$$

$$* \leq \|\mathsf{D}_\rho g\|_{\mathcal{F}L^1} \|\sigma\|_{A_p'} \|\mathsf{St}_\rho g\|_1 \|f\|_p = \|\sigma\|_{A_p'} \|f\|_p = \|\!|T|\!\|_{L^p \to L^p} \|f\|_p. \tag{54}$$

This estimate can also be reformulated as

$$\|h_\rho\|_{A_p'} \leq \|\sigma\|_{A_p'}, \quad \rho \in (0, 1]. \tag{55}$$

We are going to show that for every $f \in L^p(\mathbb{R}^d)$, one has

$$\|\sigma * f - h_\rho * f\|_{L^p} \to 0, \quad \text{for } \rho \to 0. \tag{56}$$

First, we rewrite the term to be estimated:

$$\sigma * f - h_\rho * f = \sigma * f - [(D_\rho \tau \cdot \sigma) * St_\rho g] * f = \sigma * f - (D_\rho \tau \cdot \sigma) * (St_\rho g * f). \tag{57}$$

Taking the norm on both sides and splitting the difference into two parts, we arrive at

$$I + II := \|\sigma * f - \sigma * (St_\rho g * f)\|_{L^p} + \|(\sigma - D_\rho \tau \cdot \sigma) * (St_\rho * f)\|_{L^p}. \tag{58}$$

For the first term, we have

$$I = \|\sigma * f - \sigma * (St_\rho g * f)\|_{L^p} \leq \|\sigma\|_{A_p'} \|f - St_\rho g * f\|_{L^p} < \varepsilon/2, \tag{59}$$

Hence, for some $\rho_1$, one finds that (59) is valid for $\rho \in (0, \rho_1)$.

The second term can be estimated with the help of our technical lemma, observing that the set $\{St_\eta g * f, \eta \in (0, 1]\}$ is a compact subset of $(L^p(\mathbb{R}^d), \|\cdot\|_p)$. We are thus in a situation to conclude the argument for (56) by first choosing $\rho_2 \in (0, 1]$ such that, according to Remark 14, following Lemma 4 implies that

$$\|(\sigma - D_\rho \tau \cdot \sigma) * (St_\eta * f)\|_{L^p} \leq \varepsilon/2 \text{ for } \rho \in (0, \rho_2], \eta \in (0, 1]. \tag{60}$$

By choosing $\rho_0 = \min(\rho_1, \rho_2)$ we can thus guarantee $I + II < \varepsilon$ for $\rho \in (0, \rho_0)$.

Taking the scalar product against the given $h \in L^q(\mathbb{R}^d)$, we obtain

$$\lim_{\rho \to 0} \langle T(f) - h_\rho * f, h \rangle = 0, \tag{61}$$

thus completing the proof. □

**Remark 15.** *One can compare the ultra-weak convergence of a bounded net of operators with the $w^*-$convergence of the corresponding kernels of the operators, in the sense of the Banach–Gelfand triple of operators with distributional kernels in $(S_0, L^2, S_0')(\mathbb{R}^{2d})$. Giving details here would take too much space and should be the subject of subsequent publications.*

As an immediate consequence of the above theorem, we have

**Corollary 6.** $CV_p(\mathbb{R}^d) = PM_p(\mathbb{R}^d) \equiv A_p'(\mathbb{R}^d).$

**Proof.** Let us recall that $H_G(L^p)$ consists of the operators that commute with translation. Obviously, convolution operators by ordinary functions (e.g., from $C_c(\mathbb{R}^d)$, $S_0(\mathbb{R}^d)$ or even bounded measures) have this property. Now, we assume that we have a net of such convolution operators $(T_\alpha)_r \alpha \in I$ with ultra-weak limit $T_0$. Then, it is clear that the identity

$$\langle (T_\alpha T_x)f - (T_x T_\alpha)f, h \rangle, \quad \alpha \in I, \forall f, h \in S_0(G). \tag{62}$$

implies that the same relationship is valid for the limit $T_0$, which implies that $T_0 \in H_G(L^p(G))$. This implies that $PM_p(G) \subseteq H_G(L^p(G))$. □

The proof then shows the following corollary (see (60)):

**Corollary 7.** *The subspace of convolution operators $f \mapsto h_\rho * f$, with $h_\rho \in S_0(\mathbb{R}^d)$, is dense in $H_G(L^p(\mathbb{R}^d))$ in the strong operator topology.*

## 8. Tempered Elements in $\left(L^p(G), \|\cdot\|_p\right)$

Let us conclude this paper with a hint to two related questions, concerning the algebra of *tempered elements* in $L^p(\mathbb{R}^d)$, which also has the same multiplier algebra as $\left(L^p(\mathbb{R}^d), \|\cdot\|_p\right)$. Relevant references are the author's paper [37] and a series of papers by K. McKennon (and coauthors) such as [38–42].

Essentially, the results of these papers imply (for the LCA case) that the multiplier space $H_G(L^p(G))$ coincides with the space of multipliers of $L_p^t(G) := L^p(G) \cap CV_p(G)$ (defined appropriately). Elements of this space (defined properly, without reference to the pointwise existence of convolution integrals) are called *tempered elements of* $L^p(G)$.

This raises another question (which we cannot answer at the moment): what can be said about the multipliers of $\boldsymbol{B}_p := \boldsymbol{L}^p(G) \cap A_p(G)$, with its natural norm? We have clearly

$$H_G(\boldsymbol{L}^p(G)) \hookrightarrow H_G(\boldsymbol{B}_p). \tag{63}$$

However, is this inclusion a proper one?

## 9. Conclusions and Summary

In this paper, we have shown that the identification of the multiplier space of the Herz–Figa-Talamanca space with its dual space can be derived easily by direct methods. We provide arguments showing that this multiplier algebra coincides with that of $\left(\boldsymbol{L}^p(G), \|\cdot\|_p\right)$. The setting of mild distributions helps to avoid the cumbersome setting of quasimeasures. It behaves better under the Fourier transform and allows views that are closer to that of engineers. The ultra-weak approximation of convolutors on $\left(\boldsymbol{L}^p(G), \|\cdot\|_p\right)$ can be shown to follow by means of standard regularization arguments for mild distributions.

We should also mention that our methods are more concrete than the very general approach provided by M. Rieffel ([43]), which uses more abstract concepts from the theory of Banach modules.

The approach taken also immediately extends to the context of Lorentz spaces (see [44–46]) or Orlicz spaces, as treated in recent papers [47–49].

**Funding:** This research received no external funding.

**Data Availability Statement:** Not applicable.

**Acknowledgments:** The author thanks the reviewers for going through the manuscript and improving the presentation.

**Conflicts of Interest:** The authors declare no conflict of interest.

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
