# Peer review of "A Characterization of Multipliers of the Herz Algebra"

_axioms, doi:10.3390/axioms12050482_

Round 1
Reviewer 1 Report
I have read the paper carefully. The paper is technically sound, the proofs are clearly presented. The topic of this paper is very for Algebra readers.
The key contribution of this paper is the demonstration that the space of multipliers of Ap(G) coincides with the space of multipliers of Lp(G).
The authors concluded their paper with a nice future direction.
The section literature should be changed with References.
I recommend the paper in its present form in this journal.
Author Response
AUTHOR COMMENTS:
References [1] and [10] have been replaced by the correct items
(errors caused by typos).
The terms pseudo-measure and quasi-measure have been replaced by
pseudomeasure and quasimeasure, following Derighetti.
Reviewer 2 Report
1. The authors have to compare their results with the other literature and present the advantages.
2. What is the novelty of the proposed work? Please list the motivation and contribution in the introduction section.
3. We suggest adding the conclusion part in proportion to the importance of the topic.
Author Response
Reply to Reviewer 2:
AUTHOR COMMENTS:
I have done one more spell-checking and have tried to improve the presentation of the material (this is always possible). The changes have been marked in blue. I do not know what the suggestion to ``improve the research design'' means. As an experienced author I have never heard of such a vague suggestion.
Also the suggestion to apply a spell-checker was not helpful, because I have done this more than once and could not find simple typos by running it once more. In fact, a colleague of mine has done a careful checking. Concrete suggestions could have been helpful. On the other hand, I want once more through the text and have done a couple of minor reformulations. Also, the question, of whether the conclusions are supported by the results has been answered critically. Another reviewer declared this question "not applicable".
I have made an attempt to explain this a bit more, but due to a lack of concrete comments, this was also not easy.
All the changes have been marked in the PDF in BLUE in the PDF of the REVISION.
The many smaller changes are marked in BLUE in the revision.
At the end, a short section called CONCLUSION and SUMMARY has been added.
It contains a new reference to the work of M.Rieffel (from 1967).
References [1] and [10] have been replaced by the correct items
(errors caused by typos).
The terms pseudo-measure and quasi-measure have been replaced by
pseudomeasure and quasimeasure, following Derighetti.
Reviewer 3 Report
It is difficult to do a complete review in the short turnaround time allowed by this journal. With that said I have reviewed the paper, though not at a deep level, and it appears to contain interesting and significant results. Aside from a few minor grammatical issues the paper is clearly written.
Author Response
Reply to Reviewer 3:
Suggestions for Authors
It is difficult to do a complete review in the short turnaround time allowed by this journal. With that said I have reviewed the paper, though not at a deep level, and it appears to contain interesting and significant results. Aside from a few minor grammatical issues the paper is clearly written.
AUTHOR COMMENTS:
Thanks, I understand. I have done various local improvements and
have added comments pointing out the relevance and purpose of the paper.
References [1] and [10] have been replaced by the correct items
(errors caused by typos).
The terms pseudo-measure and quasi-measure have been replaced by
pseudomeasure and quasimeasure, following Derighetti.